# Altermagnetic lifting of Kramers spin degeneracy

J. Krempaský[1,11]✉, L. Šmejkal[2,3,11], S. W. D'Souza[4,11], M. Hajlaoui[5], G. Springholz[5], K. Uhlířová[6], F. Alarab[1], P. C. Constantinou[1], V. Strocov[1], D. Usanov[1], W. R. Pudelko[1,7], R. González-Hernández[8], A. Birk Hellenes[2], Z. Jansa[4], H. Reichlová[3], Z. Šobáň[3], R. D. Gonzalez Betancourt[3], P. Wadley[9], J. Sinova[2,3], D. Kriegner[3], J. Minár[4]✉, J. H. Dil[1,10] & T. Jungwirth[3,9]✉

Lifted Kramers spin degeneracy (LKSD) has been among the central topics of condensed-matter physics since the dawn of the band theory of solids[1,2]. It underpins established practical applications as well as current frontier research, ranging from magnetic-memory technology[3–7] to topological quantum matter[8–14]. Traditionally, LKSD has been considered to originate from two possible internal symmetry-breaking mechanisms. The first refers to time-reversal symmetry breaking by magnetization of ferromagnets and tends to be strong because of the non-relativistic exchange origin[15]. The second applies to crystals with broken inversion symmetry and tends to be comparatively weaker, as it originates from the relativistic spin–orbit coupling (SOC)[16–19]. A recent theory work based on spin-symmetry classification has identified an unconventional magnetic phase, dubbed altermagnetic[20,21], that allows for LKSD without net magnetization and inversion-symmetry breaking. Here we provide the confirmation using photoemission spectroscopy and ab initio calculations. We identify two distinct unconventional mechanisms of LKSD generated by the altermagnetic phase of centrosymmetric MnTe with vanishing net magnetization[20–23]. Our observation of the altermagnetic LKSD can have broad consequences in magnetism. It motivates exploration and exploitation of the unconventional nature of this magnetic phase in an extended family of materials, ranging from insulators and semiconductors to metals and superconductors[20,21], that have been either identified recently or perceived for many decades as conventional antiferromagnets[21,24,25].

A recently developed spin-group-symmetry classification focusing on collinear magnets and, within the hierarchy of interactions, on the strong non-relativistic exchange has identified a third elementary type of magnetic phases, in addition to the conventional ferromagnets and antiferromagnets[20,21]. The exclusively distinct spin-symmetry characteristics of this emerging third, altermagnetic class are the opposite-spin sublattices connected by a real- space rotation transformation (proper or improper and symmorphic or non-symmorphic), but not connected by a translation or inversion[20,21]. By contrast, the conventional ferromagnetic (ferrimagnetic) class has one spin lattice or opposite-spin sublattices not connected by any symmetry transformation, and the conventional antiferromagnetic class has opposite-spin sublattices connected by a real-space translation or inversion. The three classes are described by mutually exclusive non-relativistic spin-group symmetries and the classification is complete for all collinear spin arrangements on crystals[20,21].

For the case of inversion connecting the opposite-spin sublattices in conventional antiferromagnets, the Kramers spin degeneracy is protected even in the presence of the relativistic SOC[26]. For the translation connecting the opposite-spin sublattices in conventional antiferromagnets, LKSD requires both SOC and inversion-symmetry breaking in the crystal, in analogy to ordinary non-magnetic systems.

The unconventional nature of altermagnets is that the rotation symmetry connecting the opposite-spin sublattices protects an antiferromagnetic-like compensated magnetic order with a vanishing net magnetization while simultaneously enabling a ferromagnetic-like LKSD without SOC and inversion-symmetry breaking[20,21]. Here we will refer to this mechanism as 'strong' altermagnetic LKSD.

Apart from the signature antiferromagnetic-like vanishing magnetization and ferromagnetic-like strong LKSD, whose presence have been traditionally considered as mutually exclusive in one physical system, altermagnets can host a range of new phenomena that are unparalleled in either the conventional ferromagnets or antiferromagnets[20,21]. A unique property associated with the alternating sign of the spin polarization in the Brillouin zone of the altermagnet is the presence of an even number of spin-degenerate nodal surfaces crossing the

[1]Photon Science Division, Paul Scherrer Institut, Villigen, Switzerland. [2]Institut für Physik, Johannes Gutenberg-Universität Mainz, Mainz, Germany. [3]Institute of Physics, Czech Academy of Sciences, Prague, Czech Republic. [4]New Technologies Research Center, University of West Bohemia, Plzeň, Czech Republic. [5]Institute of Semiconductor and Solid State Physics, Johannes Kepler University of Linz, Linz, Austria. [6]Faculty of Mathematics and Physics, Charles University, Prague, Czech Republic. [7]Physik-Institut, Universität Zürich, Zürich, Switzerland. [8]Grupo de Investigación en Física Aplicada, Departamento de Física, Universidad del Norte, Barranquilla, Colombia. [9]School of Physics and Astronomy, University of Nottingham, Nottingham, United Kingdom. [10]Institut de Physique, École Polytechnique Fédérale de Lausanne, Lausanne, Switzerland. [11]These authors contributed equally: J. Krempaský, L. Šmejkal, S. W. D'Souza. ✉e-mail: juraj.krempasky@psi.ch; jminar@ntc.zcu.cz; jungw@fzu.cz

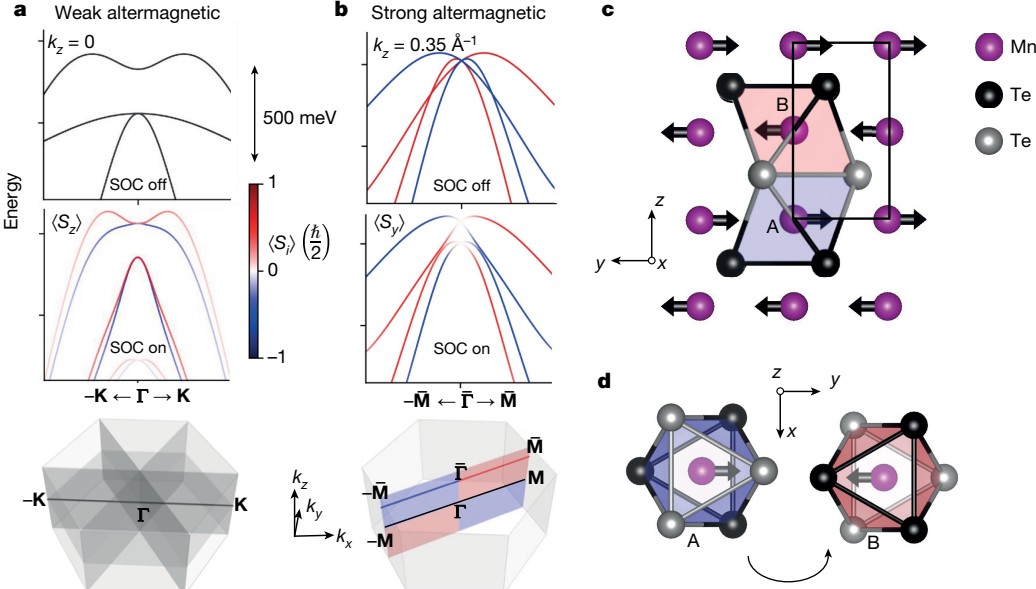

**Fig. 1 | Illustration of weak and strong altermagnetic LKSD. a**, Top and middle panels, ab initio band structure of MnTe at $k_z = 0$ along the $\mathbf{\Gamma-K}$ path for relativistic SOC turned off and on, respectively. The Néel vector is along the crystal $y$ axis (see panels **c** and **d**), corresponding to the $\mathbf{\Gamma-M}$ axis (see bottom panel of **b**). Red and blue colours correspond to opposite $z$ components of spin. Bottom panel, schematics of the Brillouin zone with four spin-degenerate nodal planes in the electronic structure with SOC turned off. **b**, Same as **a** at $k_z \approx 0.35\ \text{Å}^{-1}$ along the $\mathbf{\bar{\Gamma}-\bar{M}}$ path illustrating the strong altermagnetic LKSD.

Red and blue colours correspond to opposite $y$ components of spin. Bottom panel highlights the $\mathbf{\bar{\Gamma}-\bar{M}}$ path outside the four nodal planes (the red and blue colours highlight the alternating symmetry of the spin polarization in the plane). **c**,**d**, Schematic view of the crystal and magnetic structure of MnTe in the $y$–$z$ and $x$–$y$ planes, respectively The red and blue shadings in **c** and **d** mark Te octahedra around the Mn sites A and B with opposite spins, which are related by spin rotation combined with sixfold crystal rotation and half-unit cell translation along the $z$ axis.

zone-centre ($\mathbf{\Gamma}$ point) in the non-relativistic band structure. In Fig. 1a, we demonstrate that these spin degeneracies can be lifted by the relativistic SOC in altermagnets even without breaking the crystal inversion symmetry. We will refer to this mechanism as 'weak' altermagnetic LKSD. A comparison of the weak and strong altermagnetic LKSD is illustrated in Fig. 1a,b.

Both LKSD mechanisms can enrich fields ranging from spintronics, ultrafast magnetism, magnetoelectrics and magnonics, to topological matter, dissipationless quantum nanoelectronics and superconductivity[20,21]. For example, the strong altermagnetic LKSD has been theoretically shown to enable analogous spin-polarized currents to those used for reading and writing information in ferromagnetic memory devices while simultaneously removing the capacity and speed limitations imposed by a net magnetization[20,21,27–31]. The weak altermagnetic LKSD has been linked to Berry-phase physics governing the dissipationless anomalous Hall currents while, again, removing the roadblocks associated with magnetization for realizing robust quantum-topological variants of these effects[14,21,22,32–37]. Several of the predicted unconventional macroscopic time-reversal-symmetry-breaking responses accompanied by the vanishing magnetization have already been experimentally confirmed in altermagnetic $RuO_2$ or MnTe (refs. 22,38–41). Here, using angle-resolved photoemission spectroscopy (ARPES), we directly identify the weak and strong altermagnetic LKSD in the band structure of MnTe.

## Weak altermagnetic LKSD in MnTe

A schematic crystal structure of $\alpha$-MnTe is shown in Fig. 1c,d. The two crystal sublattices A and B of Mn atoms, whose magnetic moments order antiparallel below the transition temperature of 310 K, are connected by a non-symmorphic sixfold screw-axis rotation and are not connected by a translation or inversion[20,22]. The resulting non-relativistic electronic structure of this altermagnet is of the $g$-wave type[20] with three

spin-degenerate nodal planes parallel to the $k_z$ axis and crossing $\mathbf{\Gamma}$ and $\mathbf{K}$ points, and a fourth $k_z = 0$ nodal plane (Fig. 1a).

In Fig. 2, we show ARPES measurements[42] at 15 K using a soft X-ray photon energy of 667 eV, performed on thin MnTe(0001) films grown by molecular-beam epitaxy on a single-crystal InP(111)A substrate[22,43] (see Methods and Supplementary Figs. 1 and 2). The measurements are within the $k_z = 0$ nodal plane along $k_x$ ($\mathbf{\Gamma-K}$ path) and $k_y$ ($\mathbf{\Gamma-M}$ path). Figure 2a shows the measured raw data along the $k_x$ axis (bottom panel), compared with one-step ARPES simulation[44,45] (top panel) considering the bulk MnTe electronic structure for the initial states (see Methods). The intense spectral weight around −3.5 eV binding energy, indicated by a dashed magenta line in the experimental and theoretical panels of Fig. 2a, corresponds to a resonance resulting from Mn d states. For a better visualization of the bulk electronic structure of MnTe, this spectral weight is filtered out in the experimental ARPES band maps shown in Fig. 2b,c. Refinements by the curvature mapping[46] extracted from the area highlighted by a dashed white rectangle are shown in the insets of the top panels of Fig. 2b,c. These are compared with the corresponding relativistic ab initio electronic-structure calculations plotted in the bottom panels of Fig. 2b,c. The theoretical bands, with red and blue colours depicting opposite spin polarizations along the $z$ axis, show the weak altermagnetic LKSD within the $k_z = 0$ nodal plane. The relativistic band-structure calculations were performed assuming the Néel vector along the in-plane $y$ axis (corresponding to the $\mathbf{\Gamma-M}$ axis), consistent with earlier magnetic and magnetotransport measurements of the Néel-vector easy axis in epitaxial thin films of MnTe (refs. 22,47). Altermagnetism and SOC thus generate in this case an unconventional spin polarization of bands that is orthogonal to the direction of the magnetic-order vector. Note that the exclusive spin-polarization component along the $z$ axis of electronic states in the $k_z = 0$ plane is protected by a relativistic (non-symmorphic) mirror symmetry of the magnetic crystal.

The experimental ARPES band maps in Fig. 2b,c are fully consistent with the ab initio band structures. This includes the overall band

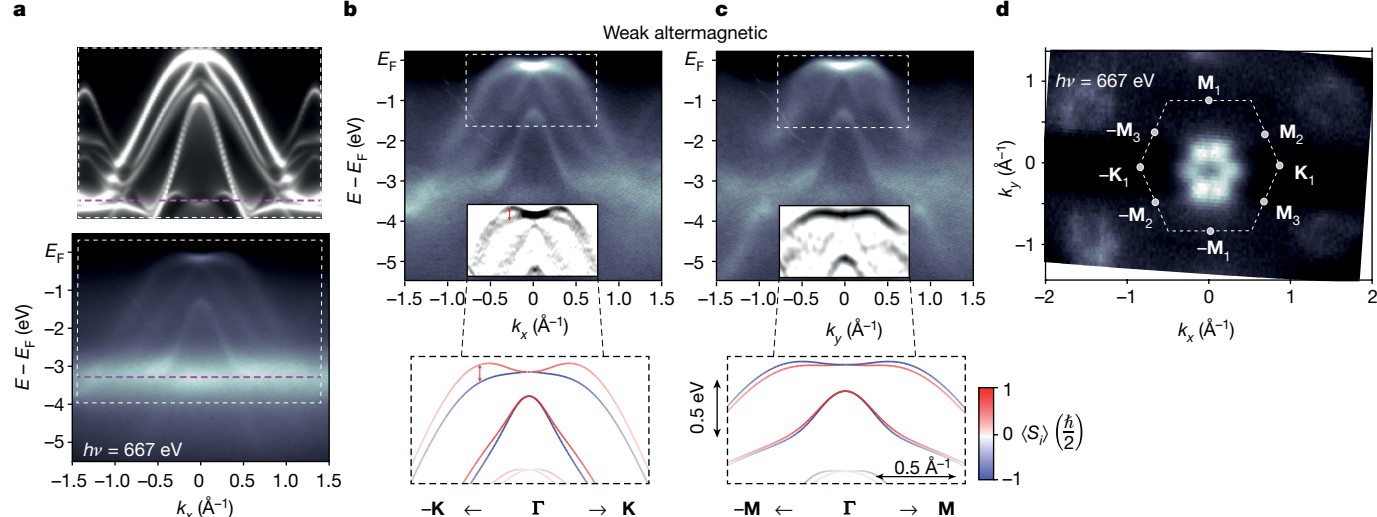

**Fig. 2 | Weak altermagnetic LKSD in the nodal plane. a**, Bottom panel, measured soft X-ray (667 eV) ARPES band map at $k_z = 0$ along $k_x$ ($\Gamma$–$K$ path) on epitaxial thin-film MnTe. Top panel, corresponding one-step ARPES simulation. Dashed magenta line highlights an intense spectral weight around −3.5 eV binding energy corresponding to a resonance of Mn d states. **b**, Measured ARPES band map along $k_x$ ($\Gamma$–$K_1$ path) after filtering out the intense spectral weight owing to the Mn d-state resonance. Inset, refinement of the measured data by curvature mapping. Bottom panel, ab initio bands, with red and blue colours corresponding to opposite $z$ components of spin. The Néel vector is aligned along the $\Gamma$–$M_2$ direction in the calculations. **c**, Same as **b** along $k_y$ ($\Gamma$–$M_1$ path). **d**, Constant-energy map obtained by integrating the measured data over a 50-meV binding-energy interval from the top of the valence band. All soft X-ray ARPES experiments were performed at 15 K.

dispersions, as well as the substantially larger splitting of the top two bands along the $k_x$ axis ($\Gamma$–$K$ path; Fig. 2b) than along the $k_y$ axis ($\Gamma$–$M$ path; Fig. 2c). The splitting is highlighted in Fig. 2b by the red double arrow in the experimental curvature map and the two split bands have opposite spins in the corresponding ab initio band structure. The approximately 100-meV scale is comparable with the record values of relativistic spin splittings in non-centrosymmetric heavy-element crystals, such as BiTeI (ref. 48). The extraordinary quadratic band dispersion and spin splitting around the $\Gamma$ point (see also Fig. 3c), consistently observed in experiment and theory, further highlight the unconventional nature of this relativistic LKSD in altermagnetic MnTe. The even-in-momentum spin splitting reflects the inversion symmetry of the altermagnetic crystal. Moreover, the lowest even spin-splitting term we observe is quadratic, whereas the constant term, and—correspondingly—the spin splitting at the $\Gamma$ point, vanish. This is consistent with earlier observations that a relativistic net magnetization in MnTe owing to canting of the sublattice moments towards the $z$ axis, allowed by the relativistic symmetry for the easy-axis orientation of the Néel vector, is extremely small[22]. It was estimated to be less than $2 \times 10^{-4}$ $\mu_B$ per Mn from ab initio calculations and remained experimentally undetectable in the thin-film MnTe epilayers[22]. The vanishingly small relativistic net magnetization, nevertheless, does not prohibit the extraordinarily large magnitude of the relativistic spin splitting away from the $\Gamma$ point in the MnTe altermagnet. The absence of the constant and linear spin-splitting terms highlights the principal distinction of this altermagnetic mechanism of LKSD from the conventional ferromagnetic-like mechanism owing to a net magnetization or the relativistic mechanism in crystals with broken inversion symmetry. In Supplementary Fig. 3, we further corroborate the altermagnetic mechanism by comparing measurements at low temperature with room-temperature measurements enabled by ultraviolet (UV) ARPES. Note that soft X-ray ARPES measurements are not realistic around or above room temperature (MnTe Néel temperature). The reason is that the Debye–Waller factor will substantially increase the incoherent spectral weight and wash out all features, resulting in a measurement of the momentum-integrated density of states[49].

Figure 2d shows a $k_z = 0$ constant-energy map measured at the soft X-ray photon energy of 667 eV, obtained by integrating the measured

data over a 50-meV interval of binding energies from the top of the valence band. The observed sixfold symmetry indicates that, within the probing area of this soft X-ray ARPES measurement ($30 \times 70$ μm$^2$), there is a comparable population of three Néel-vector easy axes, corresponding to the $\Gamma$–$M_{1-3}$ axes, which are crystallographically equivalent in the ideal hexagonal lattice of MnTe. Our observation of a multidomain state is consistent with earlier magnetotransport measurements of the MnTe epilayers[22,43]. We point out that domains with all these three Néel-vector easy axes exhibit larger spin splitting along $\Gamma$–$K_{1-3}$ paths than along $\Gamma$–$M_{1-3}$ paths, as shown in Supplementary Fig. 4. Therefore, even when the population of the three domains is comparable within the sample probing area (X-ray spot position), a substantially larger splitting is expected for the $\Gamma$–$K_{1-3}$ paths than for the $\Gamma$–$M_{1-3}$ paths. This corroborates the excellent agreement between the experimentally observed and the calculated band splittings in Fig. 2b,c.

The top-left panel of Fig. 3a shows the refinement by the curvature mapping corresponding to Fig. 2d. Together with the one-step ARPES simulation assuming an equal population of the three easy axes, shown in the top-right panel of Fig. 3a, it confirms the sixfold symmetry of this constant-energy cut. In the series of panels in Fig. 3a, we then systematically explore the symmetry of the constant-energy maps measured and calculated at different binding energies, indicated by symbols A–D in the band dispersion shown in Fig. 3c. An analogous set of measurements and calculations is shown in Fig. 3b for a different probing area on the sample (different X-ray spot position; see also Supplementary Fig. 5). Although the maps in Fig. 3a show the sixfold symmetry for all binding energies, the maps in Fig. 3b have a lower twofold symmetry at energies near the top of the valence band (binding energies A–C). The sixfold symmetry is observed in Fig. 3b only deeper in the valence band (binding energy D). The one-step ARPES simulations in Fig. 3b were performed assuming a single-domain state with the Néel vector along the easy axis corresponding to the $\Gamma$–$M_1$ axis. The agreement between experiment and theory for all of the studied constant-energy maps confirms that, in the probing area of the MnTe epilayer corresponding to Fig. 3b, there is a prevailing population of one of the three Néel-vector easy-axis domains ($\Gamma$–$M_1$ axis). Note that, in Fig. 3b, the more prominent lowering of the symmetry from

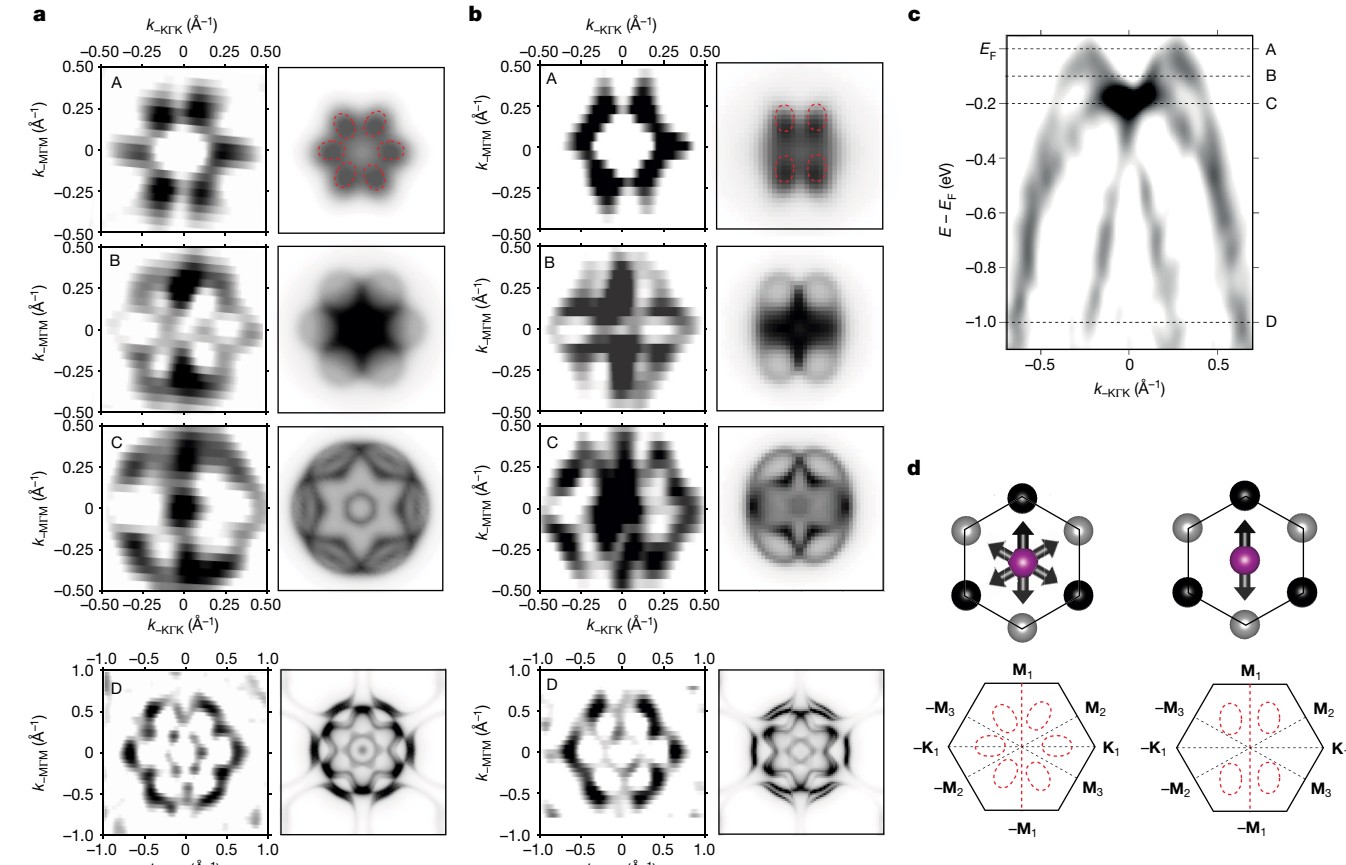

**Fig. 3 | Constant-energy maps and Néel-vector easy-axis domains. a,** Left column, refinements by the curvature mapping of measured constant-energy maps for binding energies A–D indicated in panel **c**. Right column, corresponding one-step ARPES simulations. **b,** Same as **a** for a different probing area on the sample (different X-ray spot position). Dashed red contours highlight the sixfold (twofold) symmetry in the top-right panels of **a** (**b**). **c,** Refinement by the curvature mapping of the band map from the main experimental panel of Fig. 2a at $k_z = 0$ along the $\mathbf{\Gamma}-\mathbf{K}$ path with the indicated binding energies A–D. **d,** Schematics of the sixfold symmetry of constant-energy maps (bottom left) for an equal (comparable) population of the three Néel-vector easy axes (top left) and a lowered twofold symmetry of constant-energy maps (bottom right) for one of the three easy-axes domains prevailing (top right).

sixfold to twofold near the top of the valence band correlates with the dominant contribution of p orbitals of the heavy Te atoms, which markedly enhances the strength of SOC in this spectral range (see Supplementary Fig. 2).

## Strong altermagnetic LKSD

As explained in the introduction and illustrated in Fig. 1, the strong altermagnetic LKSD can be identified in the electronic structure only outside the four nodal planes that are spin-degenerate in the non-relativistic limit. In Fig. 4, we compare the measured and simulated ARPES data inside and outside the nodal planes. Soft X-ray ARPES band maps for $k_z = 0.35$ Å$^{-1}$ (X-ray photon energy of 368 eV) along a path parallel to $\mathbf{\Gamma}-\mathbf{K}$, that is, within one of the nodal planes, are shown in Fig. 4a,b. To highlight the finite $k_z$ value, we label the path as $\mathbf{\bar{\Gamma}} - \mathbf{\bar{K}}$. Data for the same $k_z$ value and a path $\mathbf{\bar{\Gamma}} - \mathbf{\bar{M}}$, that is, outside the nodal planes, are shown in Fig. 4c,d. In both experiment and theory, we observe a substantially larger band splitting in Fig. 4c,d (strong altermagnetic), reaching a half-eV scale, than Fig. 4a,b (weak altermagnetic) in the part of the spectrum labelled by $B_1$ and $B_2$. The spin-resolved one-step ARPES simulations of this part of the spectrum then suggest that a sizeable spin-polarization signal should be detectable by spin-resolved ARPES (SARPES). This applies in particular to the $\mathbf{\bar{\Gamma}} - \mathbf{\bar{M}}$ path featuring the strong altermagnetic lifting of the spin degeneracy.

The spin-resolved measurements were performed by UV SARPES[50] on bulk-crystal samples of MnTe (see Methods and Supplementary

Figs. 6–9). The consistency between the electronic structures of the MnTe thin-film and bulk-crystal samples was confirmed by soft X-ray ARPES band maps shown in Supplementary Fig. 2a,b, accompanied by the corresponding one-step ARPES simulations in Supplementary Fig. 2c. (Note that, in both the experimental and theoretical band maps, we consistently observe that the photoemission final-state effects almost completely suppress the band mapping for $\mathbf{\Gamma}_2$ and $\mathbf{\Gamma}_4$.) The coloured stripes in the experimental panels of Supplementary Fig. 2 highlight resonances owing to Te states (purple) and Mn states (yellow–green), also observed in the simulations. In Supplementary Fig. 2d,e, we accompany the ARPES data by plotting corresponding atomic-orbital projections of ab initio bands and the density of states, consistently showing that the Te p orbitals dominate the top of the valence band, whereas the spectral weight of Mn d orbitals becomes substantial below −3 eV.

UV ARPES measurements of the MnTe bulk crystal at a photon energy of 24 eV and the corresponding simulation of the constant-energy map for non-zero $k_z$ ($k_z = 0.12$ Å$^{-1}$) are shown in Fig. 4e,f. The lowered twofold symmetry confirms the prevailing population of one of the three easy-axis domains in the UV ARPES probing area ($\approx$300 μm diameter scale) of the measured bulk-crystal sample. In the UV ARPES band map along the $\mathbf{\bar{\Gamma}} - \mathbf{\bar{M}}$ path, plotted in the left panel of Fig. 4g, we identify the analogous spectral features to those labelled as $B_1$ and $B_2$ in Fig. 4c,d, whose expected spin polarization is a result of the strong altermagnetic LKSD. The spin polarization is experimentally confirmed by the UV SARPES measurements in Fig. 4g. In the middle panel of

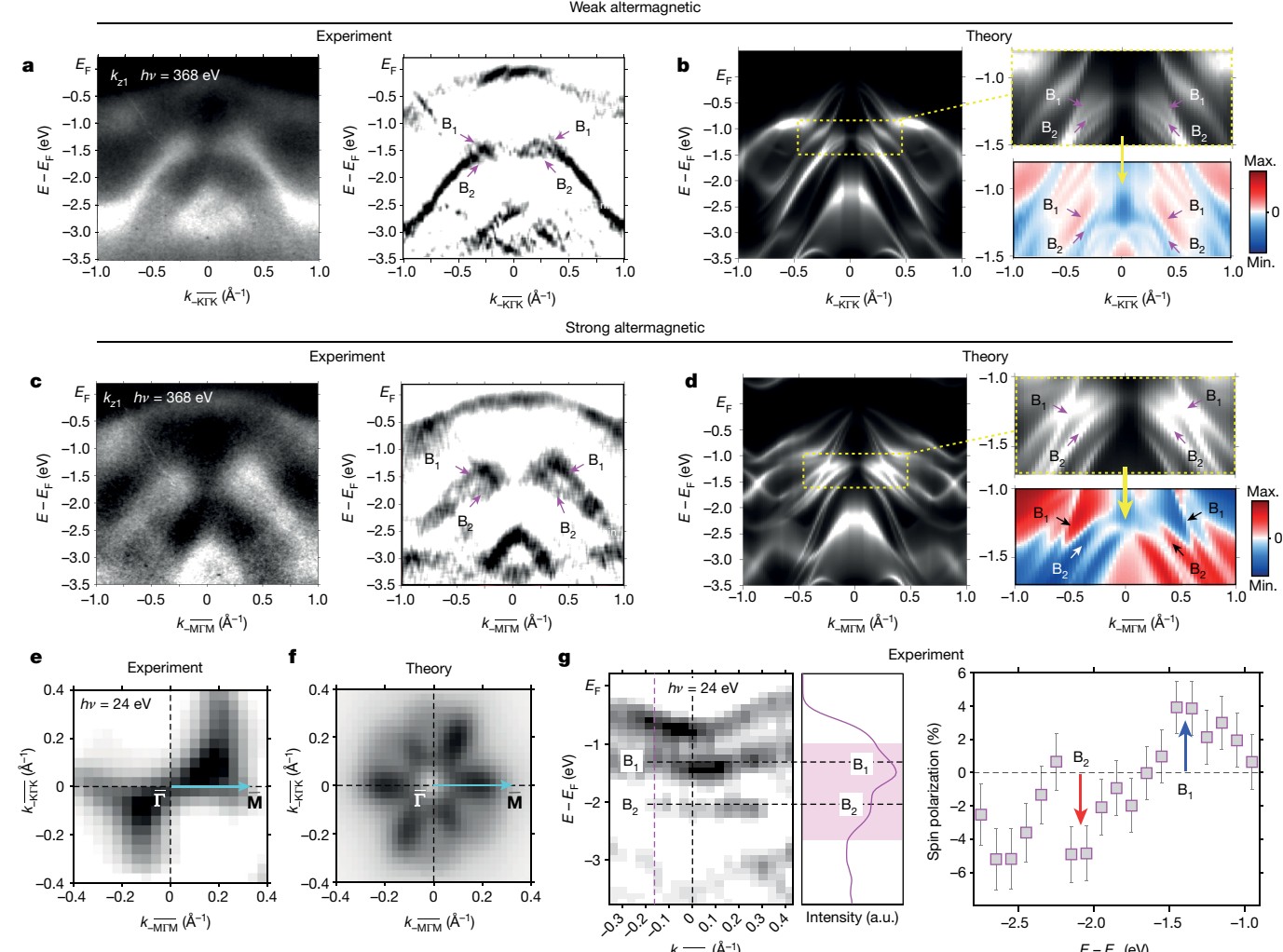

**Fig. 4 | Weak and strong LKSD at $k_z \neq 0$. a**, Measured soft X-ray (368 eV) ARPES band map at $k_z = 0.35\ \text{Å}^{-1}$ along the $\bar{\Gamma} - \bar{K}$ path (left, unrefined data; right, refined data). **b**, Corresponding one-step ARPES simulations. Red and blue colours show opposite spin polarization. **c,d**, Same as **a** and **b** along the $\bar{\Gamma} - \bar{M}$ path. **e**, Experimental UV (24 eV) ARPES constant-energy maps at $k_z = 0.12\ \text{Å}^{-1}$ measured on bulk-crystal MnTe. **f**, Corresponding one-step UV ARPES simulations. **g**, Experimental UV ARPES band map along the $\bar{\Gamma} - \bar{M}$ path (left), corresponding total-intensity energy-distribution curve (middle) and

SARPES (right). The spin-polarization component is detected along an axis corresponding to the $\bar{\Gamma} - \bar{M}$ direction (Néel-vector axis). In all theoretical panels, the considered Néel vector and the spin-polarization projection are along an axis corresponding to the $\bar{\Gamma} - \bar{M}_1$ direction (also highlighted by the cyan arrows in **e** and **f**), and the considered momentum paths are $\bar{\Gamma} - \bar{K}_1$ and $\bar{\Gamma} - \bar{M}_1$. Soft X-ray ARPES experiments were performed at 15 K, UV (S)ARPES measurements at 21 K. a.u., arbitrary units.

Fig. 4g, we plot the measured total-intensity energy-distribution curve. The right panel of Fig. 4g shows the corresponding SARPES signal for a negative value of the momentum component along the $\bar{\Gamma} - \bar{M}$ path, highlighted by a dashed purple line in the left panel of Fig. 4g. As expected, we observe the alternating sign of the spin-polarization component along the Néel vector, consistent with the presence of the strong alternmagnetic LKSD for the $\bar{\Gamma} - \bar{M}$ path (for complementary sets of consistent experimental and theoretical SARPES data, see Supplementary Figs. 10 and 11).

The agreement between the spin-split band structure observed in ARPES and that obtained from density functional theory (DFT) confirms the prediction[20,21] that alternmagnetism can originate directly from crystal symmetries, without requiring strong electronic correlations. The crystal-symmetry basis makes alternmagnetism one of the elementary phases of matter, which—remarkably—has been omitted for nearly a century of the band theory of solids. Our results highlight the strength of the spin-group-symmetry classification in unravelling new magnetic phases and in describing the hierarchy of energy scales that underpin their rich phenomenology and potential applications[20,21].

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

# Methods

### MnTe thin-film growth

MnTe(0001) epilayers of 200 nm thickness were grown by molecular-beam epitaxy on single-crystalline In-terminated InP(111) substrates using elemental Mn and Te sources. The less than 1% lattice mismatch results in single-crystalline hexagonal MnTe growth with the c axis (z axis) perpendicular to the surface. Two-dimensional growth of α-MnTe is achieved at substrate temperatures of 370–450 °C.

### MnTe thin-film characterization and sample transfer to ARPES

Details on sample characterization can be found in ref. 47. For ARPES experiments, the samples were transferred after growth into a ultrahigh-vacuum suitcase, in which they were transported to the ARPES station at the synchrotron without breaking ultrahigh-vacuum conditions.

### MnTe bulk single-crystal growth

For the growth of bulk MnTe single crystals using the self-flux method, pure manganese (99.9998%) and tellurium (99.9999%) in the molar composition $Mn_{33}Te_{67}$ were placed in an alumina (99.95%) crucible and, together with a catch crucible filled with quartz wool, sealed in a fused-silica tube under vacuum. The sample was first heated up to 1,050 °C and then cooled down to 760 °C for four days. At 760 °C, the sample was quickly put into a centrifuge, in which the crystals were separated from the remaining melt. The crystals were in forms of flat plates with lateral dimensions of several millimetres and thicknesses of hundreds of micrometres.

### MnTe bulk single-crystal characterization

For the characterization data, see Supplementary Figs. 6–9. The structural quality of MnTe bulk single crystals was confirmed by X-ray diffraction. Single-crystal X-ray-diffraction measurements were performed with a Rigaku SmartLab with 9 kW Cu rotating anode, Ge two-bounce monochromator and Hypix detector. Powder diffraction measurements for the lattice parameter determination were performed using a Panalytical Empyrean with Cu tube in Bragg–Brentano geometry. Powder diffraction simulations were performed using the xrayutilities tool for reciprocal-space conversion of scattering data[51].

Magnetization measurements by a superconducting quantum interference device (SQUID) verified the compensated magnetic ordering with the Néel temperature at 310 K and the Néel vector in the z plane, consistent with earlier reports on bulk crystals[25] and also consistent with the magnetic characteristics of our thin MnTe films. The magnetometry measurements were performed in a Quantum Design SQUID magnetometer using the reciprocating sample option for increased measurement sensitivity. Temperature-dependent susceptibility measurements were taken in a magnetic field of 50 mT.

For the X-ray diffraction and SQUID magnetometry investigations, samples were cleaned in aqua regia to remove a parasitic $MnTe_2$ phase formed at the surface during the final phases of the growth. No traces of this phase could be detected in the X-ray diffraction investigations of the single crystals after this cleaning procedure. Cleaving the MnTe platelets for ARPES measurements, therefore, also exposes a pristine α-MnTe(0001) surface.

### MnTe bulk single-crystal transfer to ARPES

Because the sample holder at the COPHEE endstation at the Swiss Light Source was not compatible with the Omicron plate used in the vacuum suitcase, and the system did not allow for decapping the MnTe surface, we performed the SARPES measurements on in situ-cleaved bulk-crystal samples.

### ARPES

ARPES was used for investigating the electronic structure of MnTe—including the Fermi surface, band structure and one-electron spectral function $A(\omega, k)$—which are resolved in electron momentum k (see ref. 52 for more details). The extension of photon energies into the soft X-ray range from a few hundred eV to approximately 2 keV reduces the cross-section of surface states compared with the UV ARPES (ref. 53) and enhances the probing depth of this technique, characterized by the photoelectron escape depth λ, by a factor of 3–5 compared with the UV ARPES (ref. 54). This enables access to the intrinsic bulk properties, which is essential for 3D materials such as MnTe. The increase of λ reduces the intrinsic broadening $\delta k_z$ of the out-of-plane momentum $k_z$, defined by the Heisenberg uncertainty principle as $\delta k_z \approx \lambda^{-1}$ (ref. 55). Combined with the free-electron dispersion of high-energy final states, the resulting precise definition of $k_z$ allows accurate determination of the 3D electronic structure. As in the case of MnTe, this advantage of soft X-ray ARPES has also been demonstrated on, for example, ferroelectric Rashba semiconductors[18], transition-metal dichalcogenides[55,56], high-fold chiral fermion systems[57] etc. The detection of bulk MnTe states in our soft X-ray ARPES measurements is confirmed in Supplementary Fig. 2 by the observed band dispersions along the $k_z$ axis, corresponding to the crystal axis normal to the film surface.

The soft X-ray ARPES experiments were conducted in the photon energy range 350–700 eV at the soft X-ray ARPES endstation[42] of the ADRESS beamline at the Swiss Light Source, Paul Scherrer Institute, Switzerland[58]. All presented data were acquired with π-polarized X-rays. The photoelectrons were detected using the PHOIBOS 150 analyser with an angular resolution of approximately 0.1° and using a deflector mode without changing the sample angles. The combined (beamline and analyser) energy resolution varied between 50 and 100 meV in the above energy range. The experiments were performed in a vacuum of better than $1 \times 10^{-10}$ mbar and at a sample temperature of around 15 K. In the presented data, the coherent spectral fraction was enhanced by subtracting the angle-integrated spectral intensity as seen in Fig. 2a–c. The constant-energy-surface maps were integrated within a range ±50 meV. The conversion of the measured photoelectron kinetic energies and emission angles to binding energies and momenta was accomplished using the kinematic formulas that account for the photon momentum[42].

The SARPES measurements were conducted at 24 eV at COPHEE experimental station at the Swiss Light Source SIS beamline[50,59] on in situ-cleaved bulk single crystals at 21 K. Combined with an angle-resolving photoelectron spectrometer, it produces complete datasets consisting of photoemission intensities (Fig. 4e), as well as spin-polarization curves (Fig. 4g) with the combined experimental resolutions of approximately 25 meV and approximately 100 meV, respectively.

### Calculations

The experimental results were compared with ab initio electronic-structure calculations, performed for bulk MnTe crystal in $P6_3/mmc$ (space group: 194) symmetry using the lattice parameter as determined from the X-ray diffraction measurements[47].

We calculated the electronic structure of MnTe in Figs. 1 and 2 with the pseudo-potential DFT code Vienna Ab initio Simulation Package (VASP)[60]. We used Perdew–Burke–Ernzerhof (PBE) + SOC + U (ref. 61), a spherically invariant type of Hubbard parameter[22] with a $8 \times 8 \times 5$ k-point grid and a 520 eV energy cutoff.

The calculations in Figs. 2, 3 and 4 were carried out using spin-polarized fully relativistic Korringa–Kohn–Rostoker (SPRKKR) Green's function method in the atomic-sphere approximation, within the rotationally invariant GGA + U scheme as implemented in the SPRKKR formalism[44,62]. The screened on-site Coulomb interaction U and exchange interaction J of Mn are set to 4.80 eV and 0.80 eV, respectively. The angular-momentum expansion of the s, p, d, f orbital wavefunctions has been used for each atom on a $28 \times 28 \times 15$ k-point grid. The energy-convergence criterion has been set to $10^{-5}$ Ry. Lloyd's formula has been used for accurate determination of the Fermi level[62–64].

The photoemission calculations for a semi-infinite surface of MnTe(0001) with Mn atoms as the termination layer at the surface were performed within the one-step model of photoemission in the spin-density-matrix formulation as implemented in the SPRKKR package[45], considering the bulk SPRKKR electronic structure for the initial states. The one-step model describes the essential physics of the photoemission process taking into account light-induced effects within the considered experimental geometry, including the photoelectron angular distribution, matrix elements and final states constructed as the time-reversed LEED states. A small, constant imaginary value of $V_i = 0.05$ eV was considered for the initial state to account for the impurity scattering. The final-state damping was described by means of a constant imaginary value of $V_i = 1.0$ eV, which has been chosen in a phenomenological way to simulate finite inelastic mean free path for excitation energies in the ARPES soft X-ray regime.

## Data availability
All data are available in the main text or the supplementary materials. Further data are available from the corresponding author on reasonable request.

## Code availability
In the manuscript, we used two ab initio DFT-based packages. The SPRKKR multiple scattering package is freely available (no costs apply) under the specific user license and the package can be downloaded following registration at https://www.ebert.cup.uni-muenchen.de/index.php/en/software-en. The Python-based interface, ase2sprkr, is published under the MIT license at https://github.com/ase2sprkr/. The Vienna Ab initio Simulation Package (VASP) used for several simulations can be purchased from https://www.vasp.at/. All input files, post-processing procedures and scripts used to evaluate experimental and theoretical data are available from the authors on request.

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

**Acknowledgements** We acknowledge fruitful discussions with K. Výborný. This work was supported by the Czech Science Foundation grant no. 19-28375X, the Ministry of Education of the Czech Republic grants CZ.02.01.01/00/22_008/0004594, LNSM-LNSpin and LM2018140 and the Neuron Endowment Fund grant. L.S. acknowledges support from the Johannes Gutenberg-Universität Mainz TopDyn initiative. J.S. acknowledge funding from the Deutsche Forschungsgemeinschaft (DFG) grant no. TRR 173 268565370 (project A03). L.S. and A.B.H. acknowledge funding from Deutsche Forschungsgemeinschaft (DFG) grant no. TRR 288 - 422213477 (Projects A09 and B05). S.W.D. and J.M. thank the QM4ST project financed by the Ministry of Education of the Czech Republic grant no. CZ.02.01.01/00/22_008/0004572, co-funded by the European Regional Development Fund. W.R.P. acknowledges support from the Swiss National Science Foundation grant no. 200021_185037. K.U. acknowledges the Ministry of Education of the Czech Republic grant no. LM2023065. D.K. acknowledges the support from the Czech Academy of Sciences (project no. LQ100102201) and the Czech Science Foundation grant no. 22-22000M. M.H. and G.S. acknowledge Austrian Science Fund grants P30960-N27 and I-4493-N. D.U. thanks the Swiss National Science Foundation for financial support within the grant 200021_197157, and F.A. for the grant 200020B-188709.

**Author contributions** J.K., L.S., S.W.D., M.H., G.S., J.M., J.H.D. and T.J. conceived the idea and proposed the experimental and modelling design. L.S., S.W.D., R.G.-H., A.B.H., Z.J., J.S., J.M. and T.J. carried out and/or analysed the band structure and ARPES simulations. M.H., G.S., K.U., H.R., Z.S., R.D.G.B., P.W. and D.K. prepared the samples and/or conducted the material characterization. J.K., M.H., F.A., P.C.C., V.S., D.U., W.R.P. and J.H.D. performed and/or analysed ARPES measurements. M.H. performed and analysed temperature-dependent ARPES measurements. J.K., L.S., J.H.D. and T.J. co-wrote the manuscript. All authors contributed to data analysis and read and commented on the manuscript.

**Competing interests** The authors declare no competing interests.

**Additional information**
**Correspondence and requests for materials** should be addressed to J. Krempaský, J. Minár or T. Jungwirth.
