## [Peer Review File · Nature]

Manuscript Title: Altermagnetic lifting of Kramers spin degeneracy

Reviewer Comments & Author Rebuttals

Reviewer Reports on the Initial Version:

Referees' comments:

Referee #1 (Remarks to the Author):

The manuscript by Krempaský and coworkers reports a detailed investigation of the electronic structure of an altermagnet MnTe and provides evidence of altermagnetic lifting of Kramers spin degeneracy by photoemission spectroscopy and ab initio calculations. The authors identified two mechanisms of the spin splittings in the altermagnet, non-relativistic strong altermagnetic lifting and relativistic weak altermagnetic lifting that needs the help of SOC. The band structures with altermagnetic splitting are experimentally mapped by ARPES, and spin-polarizations are also investigated by spin-resolved ARPES, agreeing with ab initio calculations.

The work is thoroughly performed noteworthy, as direct experimental evidence of the lifting of spin-degeneracy in altermagnets has been limited. The usage of SX-ARPES in most of the performed experiments is effective in determining the 3D band structures with its bulk sensitivity, as the effects should be disentangled from the surface effects. Moreover, the classification of the splittings presented in this manuscript is of high importance since it provides a new framework for the understanding of spin-split electronic structures. Although the "weak" altermagnetic effect is brought by SOC, the spin-texture is different from the conventional Rashba or Dresselhaus effect, demonstrating the fact that the splitting should not be captured by the existing framework. Considering the implications and the impact of the manuscript on the community, it has the potential to be published in Nature. However, there are several issues that the authors should clarify.

(1) The band splitting observed in experiments is attributed to the altermagnetic effects mainly by the agreement with DFT calculations. However, experimentally, its temperature dependence should provide strong evidence, as the altermagnetic effects are expected to appear only below the Neel temperature. In this way, the altermagnetic effects can be clearly distinguished from the known effects, such as the (hidden) Rashba/Dresselhaus effect. Can the author discuss the band structures below and above the Neel temperature to see how the splitting evolves around the magnetic phase transition?

(2) The SARPES result presented in Fig. 4g is also highly supportive of the spin splitting. However, SARPES spectra are often affected by complicated final state effects, resulting in the deviation from the spin polarizations of the initial state. I propose to add SARPES results at different momenta to check if the signs of the spin polarizations are consistent with the calculations. Additionally, it would be desirable if the authors could measure the out-of-plane spin polarizations since the weak and strong effects should induce spin polarizations in different directions.

(3) Also, the spin polarizations shown in the manuscript is very small. Is this consistent with the theoretical prediction? As the absolute value is not shown in Fig. 4b and 4d, it is not easy to compare these results quantitatively.

(4) In Ext. Data 1., the splitting is largest along the -K2-G-K2 line, which is not parallel nor perpendicular to the magnetization axis. Can the authors comment more about how to predict the magnitude of the splitting with regard to the magnetization axis?

(5) ARPES results on different spots are very illustrative in discussing the effect of AFM domains on the observed band structures. However, the relationship between the X-ray spot size and the domain size is not clear. What are the typical sizes of the domains and the light used in the

experiments? Such information will be valuable for follow-up research and future applications.

(6) While the manuscript emphasizes the large magnitude of the spin splittings, it may not be clear to the broad range of readers whether the effects are significant compared to the known effects. It may be beneficial if the authors compare the magnitude of altermagnetic splitting to those of other spin-split states in ferromagnets and Rashba systems, etc.

Referee #2 (Remarks to the Author):

In this paper, the authors provide a study of the electronic and magnetic properties of the (001) surface of MnTe. Due to the recent discovery of the altermagnetic properties of MnTe, this is a very relevant topic at the moment. At the moment, on arxiv other papers on the same topic are present. Some of the authors are the leaders of this new field named altermagnetism. Even if there are previous papers on the discovery of spin-splittings in antiferromagnets, the authors provided in the last 2 years a deep and elegant explanation of the altermagnetism and its consequences. Despite my admiration for the authors, I regret to inform you that in my humble opinion, the present paper is very far from being suitable for publication in Nature. The main issue is reported below at the point 1).

1.1) The bulk MnTe is an altermagnetic system. However, the (001) surface of the MnTe is not a pure altermagnetic system. Due to the A-type antiferromagnetic order, the surface layer will be spin-up and the subsurface layer will be spin-down. The surface layer and the subsurface layer are strongly inequivalent, therefore, you will have a strong difference between the spin-up surface states and spin-down surface states. We have surface spin-polarization. I did not see a mention the surface spin-polarization in the paper. To observe pure altermagnetic surface states, it is necessary to have at least a surface with zero net magnetization as it could be the lateral surfaces. However, the lateral surfaces are very difficult to access experimentally. Figures 4b and 4d show the bulk and surface states: the bulk states are altermagnetic but the surface states are not altermagnetic.

The authors show the inset where the altermagnetism is clearly visible, but most likely they put in the inset the bulk states. My guess is that the surface spin-polarization will be almost impossible to decouple with respect to the weak altermagnetism and very difficult to decouple respect to the strong altermagnetism. If you claim to study the (001) surface of MnTe, I want to see the mention to the spin-polarization in the abstract and/or conclusions. I do not see how the authors can solve this problem easily, therefore, I think that even the revised version of the this paper should go to Nature Publishing Group with a lower impact factor.

1.2) The alternative is that the authors would focus just on the bulk states of MnTe, should remove the calculations of the surface states and should prove that in their experimental data, the contribution from the surface is negligible.

I am open to other options, but I cannot recommend for publication a paper from which the community will understand that the (001) surface of MnTe is a pure altermagnet. Whatever decision the authors will make, a strong revision of the paper is necessary.

Below, I report other questions to which the authors should reply:

2) The author defines the altermagnetism as third phase.

I recognize that the authors gave an outstanding contribution to the field with the deep characterization and explanation of the altermagnetic systems. I completely agree with the author

that these antiferromagnets with non-relativistic spin-splitting should have a new name. However, I am really against the definition of altermagnetism as a third phase.

On this aspect, I have the same vision reported in L.-D. Yuan and A. Zunger, *Advanced Materials*, 2211966 (2023). My view is that the community should go towards the nomenclature of "conventional antiferromagnets" and "altermagnetic antiferromagnets (or altermagnets)". Anyway, I will not recommend for publication a paper where it is written "third phase".

3) The authors wrote "The weak altermagnetic mechanism generates ... quadratic dispersion ..." in the conclusions. Do the authors mean that the spin-orbit generates a quadratic dispersion?? I cannot easily obtain this consideration from their data.

Below, I report other optional corrections:

4) The weak altermagnetism is not very well explained. I guess that the authors are preparing another paper on the topic, however, I would like to read something more in this paper if possible.

5) I think that the titles of the figures would be better as "weak altermagnetism" and "strong altermagnetism"

Referee #3 (Remarks to the Author):

Authors of this manuscript present data documenting observation of spin band splitting in proposed alter magnet MnTe. The proposal of new type of magnetic order (altermagnetism) gained significant attention in condensed matter community. Experimental observation of such order is therefore an important development and certainly would warrant publication in *Nature*.

The Main conclusions of the manuscript were mostly based on similarities between the simulated and experimental ARPES signal. Even though the comparison is quite reasonable, it would be nice to have more definitive proof such as data above T_n and more extensive spin resolved data near E_f . Current spin resolved data is very limited and given the noise, error bars and very limited number of points it does not make a convincing case.

For more minor points.

1) The temperature of measurements should be indicated in captions or panels.

2) In Fig. 4 the experimental data and simulations should be clearly labeled to avoid confusion.

In summary, I find the findings quite promising, but the manuscript would benefit from several additional pieces of evidence to make the case watertight.

Author Rebuttals to Initial Comments:

Reply to the referee reports on the Manuscript:

“Altermagnetic lifting of the Kramers spin degeneracy” by J. Kremasky et al.

In the point-by-point response below we repeat the Reviewers’ comments, followed by our response and the description of the corresponding changes in the manuscript, highlighted in red.

Referee #1 report

Summary

The manuscript by Kremaský and coworkers reports a detailed investigation of the electronic structure of an altermagnet MnTe and provides evidence of altermagnetic lifting of Kramers spin degeneracy by photoemission spectroscopy and ab initio calculations. The authors identified two mechanisms of the spin splittings in the altermagnet, non-relativistic strong altermagnetic lifting and relativistic weak altermagnetic lifting that needs the help of SOC. The band structures with altermagnetic splitting are experimentally mapped by ARPES, and spin-polarizations are also investigated by spin-resolved ARPES, agreeing with ab initio calculations.

The work is thoroughly performed noteworthy, as direct experimental evidence of the lifting of spin-degeneracy in altermagnets has been limited. The usage of SX-ARPES in most of the performed experiments is effective in determining the 3D band structures with its bulk sensitivity, as the effects should be disentangled from the surface effects. Moreover, the classification of the splittings presented in this manuscript is of high importance since it provides a new framework for the understanding of spin-split electronic structures. Although the “weak” altermagnetic effect is brought by SOC, the spin-texture is different from the conventional Rashba or Dresselhaus effect, demonstrating the fact that the splitting should not be captured by the existing framework. Considering the implications and the impact of the manuscript on the community, it has the potential to be published in Nature. However, there are several issues that the authors should clarify.

We thank the Referee for this and the other stimulating comments which have motivated us to include additional data and clarifications in the manuscript.

Comment #1

(1) The band splitting observed in experiments is attributed to the altermagnetic effects mainly by the agreement with DFT calculations. However, experimentally, its temperature dependence should provide strong evidence, as the altermagnetic effects are expected to appear only below the Neel temperature. In this way, the altermagnetic effects can be clearly distinguished from the known effects, such as the (hidden) Rashba/Dresselhaus effect. Can the author discuss the band structures below and above the Neel temperature to see how the splitting evolves around the magnetic phase transition?

Response

To address this comment, we have included in our revised manuscript several additional clarifications and experimental results, providing further support to the central claims of our manuscript. We structure the response to this comment into the following points:

(i) We emphasize in the revised manuscript that the symmetric quadratic momentum-dependence of the splitting around the Γ -point, observed consistently in experiment and theory and highlighted in the panels from Fig. 1 and 3 replotted below, is fundamentally distinct from the conventional linear-in-momentum spin-splitting in relativistic non-centrosymmetric crystals (e.g. Rashba or Dresselhaus). The degeneracy at the Γ -point makes our observed splitting also principally distinct from the conventional ferromagnetic splitting due to the net magnetization which, to the lowest order, is momentum independent.

Panels from Figs. 1 and 3 of the main text highlighting the theoretically and experimentally observed unique nature of the momentum-dependent spin-split bands around the Γ -point in the MnTe altermagnet. Namely, the unique nature includes the quadratic spin-splitting dispersion and spin-degeneracy at the Γ -point.

To highlight these points, we have added the following paragraph in the revised main text:

“The extraordinary spin-splitting magnitude ~ 100 meV and the quadratic band dispersion and spin splitting around the Γ -point (see also Fig. 3c), consistently observed in experiment and theory, further highlight the unconventional nature of this lifting of the Kramers spin degeneracy in altermagnetic MnTe. The even-in-momentum spin-splitting reflects the inversion symmetry of the altermagnetic crystal. Moreover, the lowest even spin-splitting term we observe is quadratic while the constant term, and correspondingly the spin splitting at the Γ -point, vanish. This is consistent with earlier observations that a relativistic net magnetization in MnTe due to canting of the sublattice moments towards the z-axis, allowed by the relativistic symmetry for the easy-axis orientation of the Néel-vector, is extremely small [25]. It was estimated to be less than 2×10^{-4} μB per Mn from ab initio calculations, and remained experimentally undetectable in the thin-film MnTe epilayers [25]. The vanishingly small relativistic net magnetization, nevertheless, does not prohibit the extraordinarily large magnitude of the relativistic spin-splitting away from the Γ -point in the MnTe altermagnet. The absence of the constant and linear spin-splitting terms highlights the principal distinction of this altermagnetic mechanism of lifting the Kramers spin degeneracy from the conventional ferromagnetic-like mechanism due to a net magnetization or the relativistic mechanism in crystals with broken inversion symmetry.”

(ii) We have included in the revised Supplemental information additional ARPES measurements comparing low-temperature and high-temperature band structures, providing further supporting evidence for the observed altermagnetic splitting, as suggested by the Referee. The corresponding UV ARPES measurements are reproduced in the figure below, showing at low temperature the top two split altermagnetic bands, as seen in the soft X-ray ARPES data presented in the main text, while at room temperature the splitting is not detected. In addition, we point out in the revised manuscript that soft X-ray ARPES at or above room temperature (MnTe Néel temperature) suffers from strong broadening due to the Debye-Waller factor, which dramatically increases the incoherent spectral weight and washes out all features of interest. For this reason, room-temperature measurements were obtained only by UV ARPES.

To clarify these points, we have added the following paragraphs in the revised manuscript:

“...In Supplementary Fig. S3 we further corroborate the altermagnetic mechanism by comparing measurements at low temperature with room-temperature measurements enabled by UV ARPES. Note that soft X-ray ARPES measurements are not realistic around or above room temperature (MnTe Néel temperature). The reason is that the Debye-Waller factor will dramatically increase the incoherent spectral weight and wash out all features, resulting in a measurement of the momentum-integrated density of states [54].

Redaction

“...Soft X-ray ARPES experiments were performed at 15 K on the beamline ADRESS at the Swiss Light Source synchrotron facility [46, 47] (for details on the technique see Methods and Supplementary Fig. S1). Since our focus is on the bulk electronic structure of MnTe, the soft X-ray technique is favorable as it reduces the cross-section of surface states, compared to the conventional vacuum ultraviolet (UV) ARPES [48]. Moreover, the extension of photon energies into the soft X-ray range enhances the probing depth of this technique by a factor of 3-5 compared to UV ARPES [49]...”

Comment #2

(2) The SARPES result presented in Fig. 4g is also highly supportive of the spin splitting. However, SARPES spectra are often affected by complicated final state effects, resulting in the deviation from the spin polarizations of the initial state. I propose to add SARPES results at different momenta to check if the signs of the spin polarizations are consistent with the calculations. Additionally, it would be desirable if the authors could measure the out-of-plane spin polarizations since the weak and strong effects should induce spin polarizations in different directions.

Response

To address this comment, we have included in the revised Supplementary information additional SARPES measurements (see the figures below), and highlighted their qualitative consistency with theory. Specifically, we compare new SARPES data at positive momentum along the $\Gamma - M$ path with the originally presented data for the negative momentum, as well as SARPES data for all three spin components.

In the revised main text, we refer to these new figures in the following corresponding remark:

“A complementary SARPES signal for a positive value of the momentum component along the $\Gamma - M$ path, shown in Supplementary Fig. S10, is consistent with the theoretically expected prevailing antisymmetric momentum-dependence for the spin-polarization component along the Néel vector for the considered in-plane momentum path at $k_z \neq 0$. The qualitative consistency between theoretical and experimental symmetries of all three components of the spin polarization is further illustrated in Supplementary Fig. S11.”

Additional Supplementary Figs. S10 and S11, showing complementary calculations and SARPES data:

Fig.S 10. **Spin resolved ARPES simulations and measurements.** (a) One-step SARPES simulations. Red and blue colors show opposite spin-polarization components along the Néel vector, corresponding to the $\Gamma - M$ axis. (b) SARPES signals for positive and negative values of the momentum component along the $\bar{\Gamma} - \bar{M}$ path, highlighted by dashed lines in (a). (Data for the negative component are replotted from Fig. 4g of the main text.) The spin polarization component is detected along an axis corresponding to the $\Gamma - M$ direction.

Fig. S 11. **Spin resolved ARPES simulations (top) and measurements (bottom) for the three orthogonal spin-polarization components.** Consistent with the one-step SARPES simulations, the measured SARPES data show a prevailing antisymmetric dependence on momentum of the in-plane spin-polarization components (left and middle panels) along the selected momentum-distribution curve (MDC), corresponding to the dashed frame in the theoretical panels. The out-of-plane spin-polarization component (right panel) shows a prevailing symmetric dependence, again consistent with the simulations.

Comment #3

(3) Also, the spin polarizations shown in the manuscript is very small. Is this consistent with the theoretical prediction? As the absolute value is not shown in Fig. 4b and 4d, it is not easy to compare these results quantitatively.

Response

Measuring the spin polarization of bulk bands is much more complicated than for surface or 2D states. Over the last decade, we have gained experience in this, but it still poses a challenge and this might be the reason that only very few groups attempt such measurements. One of the reasons is that, in the UV range, the peak to background ratio is typically rather low. As a result, the measured spin polarization will tend to be significantly lower than in theory. Therefore, at the present stage, we focus on the qualitative comparison of the symmetry of the spin-polarized signals.

In the revised main text we have included the following corresponding remark:

“Note that we do not attempt to make a quantitative comparison of the experimental spin polarization to theory. This is because of the typically low peak to background ration for bulk bands in UV ARPES.”

Comment #4

(4) In Ext. Data 1., the splitting is largest along the -K2-G-K2 line, which is not parallel nor perpendicular to the magnetization axis. Can the authors comment more about how to predict the magnitude of the splitting with regard to the magnetization axis?

Response

Quantitatively, numerical *ab initio* theory has to be employed to predict the momentum-dependent spin-splitting magnitudes. However, symmetry provides an important guidance.

For example, in the revised caption of Supplementary Fig. S4, we have added the following remark:

“There are two $\Gamma - M_i(K_i)$ paths showing the same splitting magnitude and one with a different magnitude. This follows from symmetry when the Néel vector is along the high-symmetry easy axis corresponding to one of the $\Gamma - M_i$ directions.”

Regarding symmetry of the spin polarization, we have also included in the revised main text the following remark in the discussion of the weak alternating spin splitting:

“Note that the exclusive spin-polarization component along the z-axis of electronic states in the $k_z = 0$ plane is protected by a relativistic (non-symmorphic) mirror symmetry of the magnetic crystal.”

Comment #5

(5) ARPES results on different spots are very illustrative in discussing the effect of AFM domains on the observed band structures. However, the relationship between the X-ray spot size and the domain size is not clear. What are the typical sizes of the domains and the light used in the experiments? Such information will be valuable for follow-up research and future applications.

Response

We have included in the revised main text the probing areas in the ARPES measurements in the following remarks:

“...the probing area of this soft X-ray ARPES measurement ($30 \times 70 \mu\text{m}^2$)...”

“...the UV ARPES probing area ($\approx 300 \mu\text{m}$ diameter-scale)...”

As mentioned in the manuscript, the observed lowering of the symmetry from 6-fold to 2-fold in the measured constant-energy maps, and the agreement between measured and calculated maps over a broad binding-energy range, provide a direct spectroscopic evidence of a prevailing population of one of the three Néel-vector easy-axis domains in selected probing areas of the thin film and in the

bulk MnTe samples. Here we note that this observation is consistent with our preliminary on-going parallel study in which we experimentally map the domain structure in MnTe thin-films by X-ray magnetic circular/linear dichroism photo-electron emission microscopy (XMCD/XMLD-PEEM). We observe domain sizes on scales from $\sim 0.1 - 10 \mu\text{m}$. Since the measurements are still on-going and we depending on the allocation of corresponding beam-times, these microscopy measurements are beyond the scope of the present work and will be reported elsewhere. We also note that our preliminary microscopy measurements of the MnTe films, as well as earlier studies of thin-film antiferromagnets, indicate that the formation of multi-domains states in these compensated magnets is largely driven by defects or strain originating from the film-substrate interface. Bulk single-crystal samples are free from these defects/strains and tend to have significantly larger domains.

Comment #6

(6) While the manuscript emphasizes the large magnitude of the spin splittings, it may not be clear to the broad range of readers whether the effects are significant compared to the known effects. It may be beneficial if the authors compare the magnitude of altermagnetic splitting to those of other spin-split states in ferromagnets and Rashba systems, etc.

Response

We have addressed this comment in the revised concluding paragraph of the main text by including the relevant information as follows:

“In conclusion, we have observed two types of unconventional lifting of Kramers spin degeneracy in centrosymmetric magnetically-compensated MnTe. The weak altermagnetic mechanism generates extraordinary relativistic spin-splitting with a quadratic dispersion around the Γ -point and magnitude $\sim 100 \text{ meV}$. The scale is comparable to the record values of relativistic spin splittings in non-centrosymmetric heavy-element crystals, such as BiTeI [56]. For the strong non-relativistic mechanism we detect spin splitting that reaches a remarkable half-eV scale in the MnTe altermagnet with a net magnetization $\leq 2 \times 10^{-4} \mu_B$ per Mn. For comparison, the characteristic spin-splitting in the band structure of ferromagnetic Fe with magnetization of $4.9 \mu_B$ per Fe reaches $\approx 1-2 \text{ eV}$ scale [57].

Referee #2 report

Summary

In this paper, the authors provide a study of the electronic and magnetic properties of the (001) surface of MnTe. Due to the recent discovery of the altermagnetic properties of MnTe, this is a very relevant topic at the moment. At the moment, on arxiv other papers on the same topic are present. Some of the authors are the leaders of this new field named altermagnetism. Even if there are previous papers on the discovery of spin-splittings in antiferromagnets, the authors provided in the last 2 years a deep and elegant explanation of the altermagnetism and its consequences. Despite my admiration for the authors, I regret to inform you that in my humble opinion, the present paper is very far from being suitable for publication in Nature. The main issue is reported below at the point 1).

Comment #1

1.1) The bulk MnTe is an altermagnetic system. However, the (001) surface of the MnTe is not a pure

altermagnetic system. Due to the A-type antiferromagnetic order, the surface layer will be spin-up and the subsurface layer will be spin-down. The surface layer and the subsurface layer are strongly inequivalent, therefore, you will have a strong difference between the spin-up surface states and spin-down surface states. We have surface spin-polarization. I did not see a mention the surface spin-polarization in the paper. To observe pure altermagnetic surface states, it is necessary to have at least a surface with zero net magnetization as it could be the lateral surfaces. However, the lateral surfaces are very difficult to access experimentally.

Figures 4b and 4d show the bulk and surface states: the bulk states are altermagnetic but the surface states are not altermagnetic.

The authors show the inset where the altermagnetism is clearly visible, but most likely they put in the inset the bulk states. My guess is that the surface spin-polarization will be almost impossible to decouple with respect to the weak altermagnetism and very difficult to decouple respect to the strong altermagnetism. If you claim to study the (001) surface of MnTe, I want to see the mention to the spin-polarization in the abstract and/or conclusions. I do not see how the authors can solve this problem easily, therefore, I think that even the revised version of the this paper should go to Nature Publishing Group with a lower impact factor.

1.2) The alternative is that the authors would focus just on the bulk states of MnTe, should remove the calculations of the surface states and should prove that in their experimental data, the contribution from the surface is negligible.

I am open to other options, but I cannot recommend for publication a paper from which the community will understand that the (001) surface of MnTe is a pure altermagnet. Whatever decision the authors will make, a strong revision of the paper is necessary.

Response

We thank the Referee for this and the other stimulating comments which helped us to improve the clarity of our manuscript. Let us structure our response to this comment into the following points:

(i) Since the explored weak and strong altermagnetic lifting of the Kramers spin degeneracy are effects in the band structure of bulk MnTe crystal, all our theoretical figures are based on the bulk *ab initio* band structure (we have not performed any band structure calculations in the slab geometry).

This is now explicitly emphasized in the revised main text and methods as follows:

“...Korringa-Kohn-Rostoker ab-initio approach that represents the electronic structure in terms of single-particle Green’s functions [50, 51], and considers the bulk MnTe electronic structure for the initial states (for details on ab-initio calculations see Methods).”

(ii) A key signature of the altermagnetic band structure is the symmetric quadratic momentum-dependence of the splitting around the Γ -point, observed consistently in experiment and theory and highlighted in the panels from Fig. 1 and 3 replotted below. This is fundamentally distinct from the conventional linear-in-momentum spin-splitting in relativistic non-centrosymmetric crystals, including e.g. the 2D Rashba splitting at surfaces breaking the inversion symmetry. The degeneracy at the Γ -point makes our observed splitting also principally distinct from the conventional ferromagnetic splitting due to the net magnetization which, to the lowest order, is momentum independent. This includes also splitting due to the net magnetization from an uncompensated ferromagnetic 2D surface layer.

Panels from Figs. 1 and 3 of the main text highlighting the theoretically and experimentally observed unique nature of the momentum-dependent spin-split bands around the Γ -point in the MnTe altermagnet. Namely, the unique nature includes the quadratic spin-splitting dispersion and spin-degeneracy at the Γ -point.

To highlight these points, we have added the following paragraph in the revised manuscript:

“The extraordinary spin-splitting magnitude ~ 100 meV and the quadratic band dispersion and spin splitting around the Γ -point (see also Fig. 3c), consistently observed in experiment and theory, further highlight the unconventional nature of this lifting of the Kramers spin degeneracy in altermagnetic MnTe. The even-in-momentum spin-splitting reflects the inversion symmetry of the altermagnetic crystal. Moreover, the lowest even spin-splitting term we observe is quadratic while the constant term, and correspondingly the spin splitting at the Γ -point, vanish. This is consistent with earlier observations that a relativistic net magnetization in MnTe due to canting of the sublattice moments towards the z-axis, allowed by the relativistic symmetry for the easy-axis orientation of the Néel-vector, is extremely small [25]. It was estimated to be less than $2 \times 10^{-4} \mu\text{B}$ per Mn from ab initio calculations, and remained experimentally undetectable in the thin-film MnTe epilayers [25]. The vanishingly small relativistic net magnetization, nevertheless, does not prohibit the extraordinarily large magnitude of the relativistic spin-splitting away from the Γ -point in the MnTe altermagnet. The absence of the constant and linear spin-splitting terms highlights the principal distinction of this altermagnetic mechanism of lifting the Kramers spin degeneracy from the conventional ferromagnetic-like mechanism due to a net magnetization or the relativistic mechanism in crystals with broken inversion symmetry.”

(iii) The focus on the bulk electronic structure of MnTe is reflected in our choice of the soft X-ray technique as the principal experimental tool of our study because of its superior bulk sensitivity.

This is now explicitly highlighted in the main text as follows:

“Soft X-ray ARPES experiments were performed at 15 K on the beamline ADDRESS at the Swiss Light Source synchrotron facility [46, 47] (for details on the technique see Methods and Supplementary Fig. S1). Since our focus is on the bulk electronic structure of MnTe, the soft X-ray technique is favorable as it reduces the cross-section of surface states, compared to the conventional vacuum ultraviolet (UV) ARPES [48]. Moreover, the extension of photon energies into the soft X-ray range enhances the probing depth of this technique by a factor of 3-5 compared to UV ARPES [49].”

(iv) We confirm the bulk nature of the detected band structure by the experimentally observed band dispersion along the k_z -axis, which agrees with the bulk band structure calculations.

In the revised main text we emphasize this as follows:

“The detection of bulk MnTe states in our soft X- ray ARPES measurements is confirmed in Supplementary Fig. S2 by the observed band dispersions along the k_z -axis, corresponding to the crystal axis normal to the film surface.”

(v) We demonstrate the agreement between the theoretical bulk electronic structure and ARPES data on the in-plane band dispersions. The agreement includes the larger splitting observed along the $\Gamma - \mathbf{K}$ path than along the $\Gamma - \mathbf{M}$ path at $k_z=0$ (Fig. 2 of the main text), and the opposite trend for the two in-plane paths at $k_z \neq 0$ (Fig. 4 of the main text). Next, we demonstrate the agreement between the theoretical bulk electronic structure and ARPES data on the series of constant-energy maps for a broad range of binding energies (Figs. 3 of the main text). We also observe a qualitative agreement between the theoretical bulk spin polarizations and the experimental SARPES data (Fig. 4 of the main text). To further strengthen this point, we have included in the revised Supplementary information additional SARPES measurements (see the figures below), comparing positive momentum along the $\Gamma - \mathbf{M}$ path with the originally presented negative momentum, as well as all three spin components, and highlight their qualitative consistency with the theoretical bulk spin polarization.

In the revised main text, we have included the following corresponding remark:

“A complementary SARPES signal for a positive value of the momentum component along the $\Gamma - \mathbf{M}$ path, shown in Supplementary Fig. S10, is consistent with the theoretically expected prevailing antisymmetric momentum-dependence for the spin-polarization component along the Néel vector for the considered in-plane momentum path at $k_z \neq 0$. The qualitative consistency between theoretical and experimental symmetries of all three components of the spin polarization is further illustrated in Supplementary Fig. S11.”

Additional Supplementary Figs. S10 and S11, showing complementary calculations and SARPES data:

Fig. S 10. **Spin resolved ARPES simulations and measurements.** (a) On - tep SARPES simulations. Red and blue colors show oppo ite spin-polariza ion components along the eel vector corresponding o her - \mathbf{M} axis. (b) SARPES signals for positive and negative values of the momentum componen along th \mathbf{f}' - \mathbf{M} path highlighted by dashed lines in (a). (Data for the negative component are replotted from Fig. 4g of he main ext.) The spin polarization component is detected along an axis corresponding to the \mathbf{r} - \mathbf{M} direction.

Fig. S 11. **Spin resolved ARPES simulations (top) and measurements (bottom) for the three orthogonal spin-polarization components.** Consistent with the one-step SARPES simulations, the measured SARPES data show a prevailing antisymmetric dependence on momentum of the in-plane spin-polarization components (left and middle panels) along the selected momentum-distribution curve (MDC), corresponding to the dashed frame in the theoretical panels. The out-of-plane spin-polarization component (right panel) shows a prevailing symmetric dependence, again consistent with the simulations.

Referee's remark: Below, I report other questions to which the authors should reply:

Comment #2

2) The author defines the altermagnetism as third phase. I recognize that the authors gave an outstanding contribution to the field with the deep characterization and explanation of the altermagnetic systems. I completely agree with the author that these antiferromagnets with non relativistic spin-splitting should have a new name. However, I am really against the definition of altermagnetism as a third phase.

On this aspect, I have the same vision reported in L.-D. Yuan and A. Zunger, *Advanced Materials*, 2211966 (2023). My view is that the community should go towards the nomenclature of "conventional antiferromagnets" and "altermagnetic antiferromagnets (or altermagnets)". Anyway, I will not recommend for publication a paper where it is written "third phase".

Response

We understand that the nomenclature of the different magnetic phase may be debated. However, we point out that in physics, one of the most common tools for a basic classification of phases of matter is symmetry. In our paper we refer to altermagnets as a distinct phase in this sense, i.e., from the basic symmetry-classification viewpoint. We recall explicitly in our manuscript that the spin-group symmetry classification we consider in the introduction focuses, within the hierarchy of interactions, on the strong non-relativistic exchange. We also state that the non-relativistic spin-group classification we consider in the introduction is limited to collinear magnets. We then recall the characteristic non-relativistic spin-group symmetries of the three mutually exclusive symmetry classes that we call: conventional ferromagnets and antiferromagnets, and altermagnets. The spin-group classification into the three mutually exclusive symmetry classes is mathematically rigorous, systematic and complete for all collinear spin arrangements on crystals, as derived in detail in Ref. [23].

To emphasize the last point, we have included into the introduction the following remark:

“The three classes are described by mutually exclusive non-relativistic spin-group symmetries, and the classification is complete for all collinear spin arrangements on crystals [23, 24].”

Within the spin-group symmetry classification considered in our introduction, referring to three classes is, therefore, not a matter of our choice but a mathematically rigorous result that all collinear magnets fall into one of just three mutually exclusive classes.

For describing non-relativistic spin-dependent phenomena, such as the giant and tunneling magnetoresistance and spin-transfer torques (Refs. [31,34] given in our manuscript), the spin-group classification and description is a natural symmetry tool of choice. However, as elaborated on in the manuscript, our discovery of the weak altermagnetic lifting of the Kramers spin degeneracy, albeit requiring relativistic spin-orbit coupling, is also directly enabled by the spin-group symmetry description of altermagnets. It is the identification of the spin-degenerate nodal surfaces in the non-relativistic electronic structure by the spin-group symmetries that leads us to the prediction and experimental identification of the unconventional relativistic spin splitting at these nodal surfaces, that we refer to as the weak altermagnetic lifting of Kramers spin degeneracy.

We note that the non-relativistic spin-group symmetries already played a key role in earlier studies predicting and subsequently observing relativistic transport effects in altermagnets, namely the anomalous Hall effects of comparable strength to conventional ferromagnets. The spin-group symmetries identified a strong breaking of the time-reversal symmetry by the altermagnetic order in the non-relativistic electronic structure akin to ferromagnets but, unlike ferromagnets, in the absence of a net magnetization. Since it is the time-reversal symmetry breaking in the electronic structure and not the magnetization that is at the origin of the anomalous Hall effect, the spin-group symmetries determine that the anomalous Hall effect can be equally strong in altermagnets as in ferromagnets. (The relativistic spin-orbit coupling is an equally necessary additional condition for generating the anomalous Hall effect in both collinear altermagnets and ferromagnets.) These points are extensively discussed in Refs. [17,23-25,35] given in our manuscript.

Comment #3

3) The authors wrote "The weak altermagnetic mechanism generates ... quadratic dispersion ..." in the conclusions. Do the authors mean that the spin-orbit generates a quadratic dispersion?? I cannot easily obtain this consideration from their data.

Response

To clarify this point, we added the following explanation in the revised manuscript for the quadratic dispersion:

“The extraordinary spin-splitting magnitude ~ 100 meV and the quadratic band dispersion and spin splitting around the Γ -point (see also Fig. 3c), consistently observed in experiment and theory, further highlight the unconventional nature of this lifting of the Kramers spin degeneracy in altermagnetic MnTe. The even-in-momentum spin-splitting reflects the inversion symmetry of the altermagnetic crystal. Moreover, the lowest even spin-splitting term we observe is quadratic while the constant term, and correspondingly the spin splitting at the Γ -point, vanish. This is consistent with earlier observations that a relativistic net magnetization in MnTe due to canting of the sublattice moments towards the z-axis, allowed by the relativistic symmetry for the easy-axis orientation of the Néel-vector, is extremely small [25]. It was estimated to be less than 2×10^{-4} μB per Mn from ab initio calculations, and remained experimentally undetectable in the thin-film MnTe epilayers [25]. The vanishingly small relativistic net magnetization, nevertheless, does not prohibit the extraordinarily large magnitude of the relativistic spin-splitting away from the Γ -point in the MnTe altermagnet. The absence of the constant and linear spin-splitting terms highlights the principal distinction of this altermagnetic mechanism of lifting the Kramers spin degeneracy from the conventional ferromagnetic-like mechanism due to a net magnetization or the relativistic mechanism in crystals with broken inversion symmetry.”

Referee’s remark: *Below, I report other optional corrections:*

Comment #4

4) The weak altermagnetism is not very well explained. I guess that the authors are preparing another paper on the topic, however, I would like to read something more in this paper if possible.

Response

Apart from the additional explanation mentioned above in the response to Comment #3, we have also included the following remark on the z-component of the spin polarization:

“Note that the exclusive spin-polarization component along the z-axis of electronic states in the $k_z = 0$ plane is protected by a relativistic (non-symmorphic) mirror symmetry of the magnetic crystal.”

Next, in the revised caption of Supplementary Fig. S4, we have added the following remark:

“There are two $\Gamma - \mathbf{M}_i(\mathbf{K}_i)$ paths showing the same splitting magnitude and one with a different magnitude. This follows from symmetry when the Néel vector is along the high-symmetry easy axis corresponding to one of the $\Gamma - \mathbf{M}_i$ directions.”

Comment #5

5) I think that the titles of the figures would be better as "weak altermagnetism" and "strong altermagnetism"

Response

In our manuscript the term altermagnetism refers to one specific symmetry class of collinear spin arrangements on crystals (one specific magnetic-phase type) as defined by the spin-group symmetry classification. With this definition, we do not further subdivide the altermagnetic phase of MnTe. In contrast, the weak and strong altermagnetic lifting of Kramers spin degeneracy are two phenomena, generated in the altermagnetic phase of MnTe by two principally distinct mechanisms, one requiring the relativistic spin-orbit coupling and the other one not. For these reasons we prefer to keep the original titles of the figures.

Referee #3 report

Summary

Authors of this manuscript present data documenting observation of spin band splitting in proposed alter magnet MnTe. The proposal of new type of magnetic order (altermagnetism) gained significant attention in condensed matter community. Experimental observation of such order is therefore an important development and certainly would warrant publication in Nature.

Comment

The Main conclusions of the manuscript were mostly based on similarities between the simulated and experimental ARPES signal. Even though the comparison is quite reasonable, it would be nice to have more definitive proof such as data above T_n and more extensive spin resolved data near E_f . Current spin resolved data is very limited and given the noise, error bars and very limited number of points it does not make a convincing case.

Response

We thank the Referee for these two stimulating comments which have motivated us to include additional data and clarifications in the manuscript. We structure the response into the following two points:

(i) Because soft X-ray ARPES measurements around or above room temperature (MnTe Néel temperature) are not realistic experimentally, we have included in the revised Supplementary information complementary UV ARPES measurements (see the figure below) which at low temperature show the top two split bands seen in the soft X-ray ARPES data, while at room temperature the splitting is not detected.

In the main text we have included the following remarks:

“Soft X-ray ARPES experiments were performed at 15 K on the beamline ADDRESS at the Swiss Light Source synchrotron facility [46, 47] (for details on the technique see Methods and Supplementary Fig. S1). Since our focus is on the bulk electronic structure of MnTe, the soft X-ray technique is favorable as it reduces the cross-section of surface states, compared to the conventional vacuum ultraviolet (UV) ARPES [48]. Moreover, the extension of photon energies into the soft X-ray range enhances the probing depth of this technique by a factor of 3-5 compared to UV ARPES [49]...”

“...In Supplementary Fig. S3 we further corroborate the altermagnetic mechanism by comparing measurements at low temperature with room-temperature measurements enabled by UV ARPES.

Note that soft X-ray ARPES measurements are not realistic around or above room temperature (MnTe Néel temperature). The reason is that the Debye-Waller factor will dramatically increase the incoherent spectral weight and wash out all features, resulting in a measurement of the momentum-integrated density of states [54].

Additional Supplementary Fig. S3, showing UV ARPES below and above room temperature:

Fig. S 3. Comparison of low-temperature (left) and high-temperature (right) UV ARPES measurements. Measurements were performed at photon energy 21 eV corresponding to $k_z = 0$. Splitting of the top two bands at low temperature, consistent with the splitting observed by soft X-ray ARPES (Figs. 2,3 in the main text), is highlighted by white dotted lines. The experiments were performed on the UV ARPES beamline URANOS at SOLARIS synchrotron facility. Samples used in these measurements are MnTe thin films grown by molecular-beam epitaxy. We used a vacuum suitcase to transfer the thin-film samples from the growth to the UV ARPES chamber without breaking ultra-high-vacuum conditions.

(ii) To address the second part of this Referee’s comment, we have included in the revised Supplementary information additional SARPES measurements (see the figures below), comparing positive momentum along the $\Gamma - M$ path with the originally presented negative momentum, as well as all three spin components, and we highlight their qualitative consistency with theory.

In the revised main text, we have added the following corresponding remark:

“A complementary SARPES signal for a positive value of the momentum component along the $\Gamma - M$ path, shown in Supplementary Fig. S10, is consistent with the theoretically expected prevailing antisymmetric momentum-dependence for the spin-polarization component along the Néel vector for the considered in-plane momentum path at $k_z \neq 0$. The qualitative consistency between theoretical and experimental symmetries of all three components of the spin polarization is further illustrated in Supplementary Fig. S11.”

Additional Supplementary Figs. S10 and S11, showing complementary calculations and SARPES data:

Fig. S 10. **Spin resolved ARPES simulations and measurements.** (a) On - tep SARPES simulations. Red and blue colors show oppo ite spin-polariza ion components along the eel vector corresponding o her - \mathbf{M} axis. (b) SARPES signals for positive and negative values of the momentum componen along th $\mathbf{f}'\text{-}\mathbf{M}$ path highlighted by dashed lines in (a). (Data for the negative component are replotted from Fig. 4g of he main ext.) The spin polarization component is detected along an axis corresponding to the $\mathbf{r}'\text{-}\mathbf{M}$ direction.

Fig. S 11. **Spin resolved ARPES simulations (top) and measurements (bottom) for the three orthogonal spin-polarization components.** Consistent with the one-step SARPES simulations, the measured SARPES data show a prevailing antisymmetric dependence on momentum of the in-plane spin-polarization components (left and middle panels) along the selected momentum-distribution curve (MDC), corresponding to the dashed frame in the theoretical panels. The out-of-plane spin-polarization component (right panel) shows a prevailing symmetric dependence, again consistent with the simulations.

Minor comment #1

1) The temperature of measurements should be indicated in captions or panels.

Response

We now state the temperature of the measurements in the figure captions.

Minor comment #2

2) In Fig. 4 the experimental data and simulations should be clearly labeled to avoid confusion.

Response

We have included the corresponding labels in Fig. 4.

Reviewer Reports on the First Revision:

Referees' comments:

Referee #1 (Remarks to the Author):

The authors have provided new temperature-dependent results and SARPES results, making their discussion more robust. Their findings are novel and important; therefore, I would be happy to recommend the present manuscript for publication in Nature if the one comment below is properly addressed.

In Fig. S10, it is not straightforward to compare the SARPES results with the calculations since the energy position of each band is not specified in the right panel. Probably, the authors can add markers as they already did in the main Fig. 4g.

Referee #2 (Remarks to the Author):

The authors have replied to all my questions and criticism. Modifications and additional information provided are robust. They have made more clear in the main text and supp. materials that they focus on the bulk properties. I recommend the revised version of the paper for publication.

Referee #3 (Remarks to the Author):

I think that authors reasonably addressed main points of criticism from all Referees. I find the new temperature dependent data quite convincing and would encourage authors to include this plot in main text, as together with expected dispersion, this is really a smoking gun for detection of the altermagnetic state. I support publication of this manuscript.